# Progressive Semantic Fusion Transformer for Zero-Shot Temporal Action Localization

## Abstract

Zero-Shot Temporal Action Localization (ZSTAL) aims to classify and localize action instances from unseen categories in videos. Existing ZSTAL approaches predominantly rely either on visual modality alone or on stage-limited fusion of visual and textual modalities to generate proposals. Such approaches hinder text embeddings from providing semantic guidance throughout the pipeline, limiting the model's ability to capture discriminative visual features of unseen actions. To mitigate this limitation, we propose **PSFTR** (*Progressive Semantic Fusion TRansformer*), a novel transformer-based method that progressively integrates textual semantics across stages of the pipeline. Specifically, PSFTR injects textual embeddings into both the **encoder** and **decoder** stages via a cross-attention mechanism, enabling the model to focus on text-relevant visual features and generate semantically guided learnable queries. Furthermore, during the **classification** stage, we design a query enhancement mechanism driven by textual semantic prototypes to refine the representations of action moments within the learnable queries. Extensive experiments on THUMOS14 and ActivityNet1.3 demonstrate that PSFTR achieves 28.99% mAP (+1.08%) and 29.91% mAP (+1.81%), respectively, validating the effectiveness of progressive semantic fusion for ZSTAL.

## 1 Introduction

Temporal action localization (TAL) Shao et al. (2023); Li et al. (2024a) is a fundamental task in video understanding that aims to detect and localize action segments in untrimmed videos. While substantial progress has been achieved in conventional closed-set TAL Zhu et al. (2024); Kim et al. (2025a), where the action categories at test time are identical to those seen during training, this setting severely limits the scalability and applicability of TAL in open-world scenarios Zhang et al. (2020); Gupta et al. (2024). In practice, human actions exhibit an open-ended distribution, making it infeasible to annotate all possible action categories in advance. To address this challenge, zero-shot temporal action localization (ZSTAL) Yan et al. (2023); Raza et al. (2024) has gained increasing interest. ZSTAL aims to detect and localize unseen actions by transferring knowledge from seen actions. By removing the closed-set constraint and leveraging textual semantic priors, ZSTAL enhances the generalization ability of TAL, enabling effective open-world action understanding.

Existing ZSTAL methods Ju et al. (2022); Lee et al. (2024); Li et al. (2024b) are largely built upon closed-set TAL frameworks Shi et al. (2022); Yang et al. (2024); Li et al. (2025d) and can be broadly categorized into three representative paradigms: proposal-based, query-based, and anchor-free, as illustrated in Fig. 1 (a)-(c). The proposal-based paradigm Ju et al. (2022); Li et al. (2024b) follows a two-stage pipeline: candidate action proposals are generated and then aligned with textual embeddings through visual-textual similarity for classification. The query-based paradigm Nag et al. (2022); Phan et al. (2024) leverages a set of learnable queries to predict action categories and temporal boundaries. In most existing methods, textual input is used solely as a similarity metric for classification, while both the encoder and decoder are optimized exclusively on visual features. The anchor-free paradigm Lee et al. (2024) incorporates textual semantics into the anchoring process, guiding the model to focus on frames with higher action likelihood, but it still predicts action boundaries without explicit textual semantics guidance. Despite their promising generalization capability, these approaches share a critical limitation: *they struggle to capture the critical action features of unseen categories*. This limitation may arise from three primary issues: ❶ **Modalities bottleneck**. During visual feature refinement, models Phan et al. (2024); Li et al. (2024b) fail to ef-

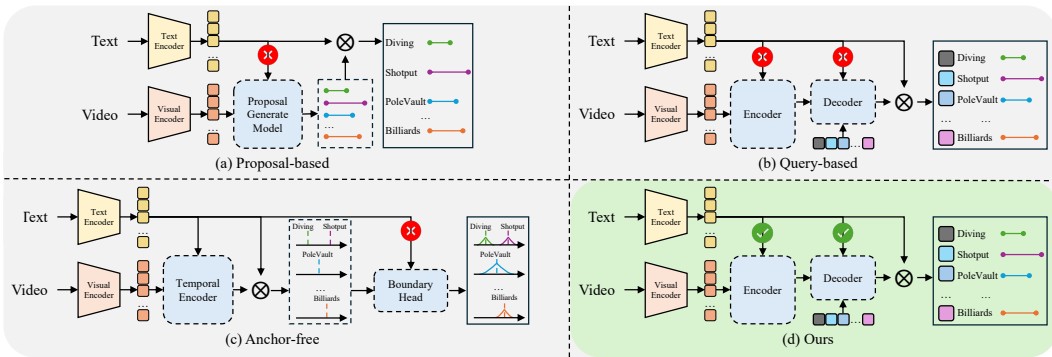

Figure 1: Overview of approaches for ZSTAL. (a) Proposal-based: generates candidate segments and then classifies them based on text-proposal similarity. (b) Query-based: employs text only in similarity computation with learnable queries. (c) Anchor-free: leverages textual guidance for frame-level processing and generates dense proposals. (d) Ours: a method that progressively integrates textual guidance into both the encoder and decoder before similarity computation.

fectively attend to action visual features aligned with textual semantics, which restricts cross-modal fusion; ❷ **Proposal bias**. Proposal generation Lee et al. (2024); Phan et al. (2024); Li et al. (2024b) is performed independently of textual semantics and relies solely on visual features, often producing action proposals that are poorly aligned with the corresponding textual semantics; ❸ **Saliency dilution**. In zero-shot settings, models Phan et al. (2024); Li et al. (2024b) typically predict action categories by comparing aggregated features with textual embeddings. This aggregation process dilutes salient visual features, weakening alignment with textual semantics. In summary, existing methods either underutilize textual semantics or restrict their use to a single stage of the pipeline. Such **stage-limited** semantic integration fails to establish coherent textual guidance across the pipeline, ultimately reducing the model's ability to capture discriminative visual features of unseen actions.

Motivated by the aforementioned issues, we propose **PSFTR** (*Progressive Semantic Fusion TRansformer*), a novel transformer-based method for ZSTAL. PSFTR progressively integrates textual semantics across critical stages of the pipeline: **feature refinement**, **query decoding**, and **query classification**, enhancing the model's ability to capture the visual characteristics of unseen actions and enabling accurate localization of these actions. Specifically, to address issue ❶, we propose the **Semantic Synergy-Aware Encoder (SSAE)** module in the encoder stage. SSAE injects textual semantics into the transformer encoder and leverages cross-modal attention to jointly extract text-relevant visual features and action-relevant textual embeddings, improving the expressiveness of both single-modality representations. To tackle issue ❷, we design the **Semantics-Guided Query Generation (SGQG)** component for the decoder stage. SGQG incorporates the textual embeddings generated by SSAE into the learnable queries. Through a deformable attention mechanism, these queries are guided to attend to visual features that are semantically aligned with the textual semantics carried by the encoder outputs. To address issue ❸, we propose the **Semantic Prototype-Driven Enhancement (SPDE)** strategy for the classification stage. SPDE filters and fuses visual features associated with textual semantics into the learnable queries generated by SGQG, enhancing the discriminative saliency of action moments within each query. By performing progressive visual-text fusion across the entire pipeline, PSFTR leverages textual semantics throughout stages, enabling it to effectively capture the characteristics of unseen actions and achieve robust ZSTAL.

In summary, our key contributions are summarized as follows:

- We propose PSFTR, a novel transformer-based method that progressively integrates semantic context into the encoder, decoder, and classifier, which facilitates the alleviation of limitations in existing ZSTAL methods via unified vision-text reasoning.
- We design three core modules: (1) SSAE aligns visual features with textual semantics to ground representations in text-relevant concepts; (2) SGQG generates semantically guided detection queries, enabling adaptive localization of novel actions; and (3) SPDE enhances the queries' classification by integrating text-relevant semantic prototypes. Together, these modules establish a coherent and semantically enriched detection pipeline for ZSTAL.

- We conduct extensive experiments on THUMOS14 and ActivityNet1.3, showing that PS-FTR outperforms the state-of-the-art (SOTA) ZSTAL methods. Comprehensive ablation studies further verify the effectiveness and contribution of each module.

## 2 RELATED WORK

**Vision-Language Models.** Vision-Language (ViL) models leverage large-scale image-text pairs to learn aligned visual and textual modalities via contrastive learning, enabling strong generalization to unseen categories. Representative models such as CLIP Radford et al. (2021) and ALIGN Jia et al. (2021) are trained on hundreds of millions of image-text examples using transformer backbones, producing powerful pre-trained encoders for various downstream tasks Li et al. (2025b); Xi et al. (2025); Yu et al. (2025); Kim et al. (2025b). To adapt ViL models to specific domains, efficient fine-tuning techniques such as adapters Gao et al. (2024); Murugesan et al. (2024) and prompt tuning Zhou et al. (2022a;b); Khattak et al. (2023) have been introduced to inject task-specific knowledge while retaining their generalization ability. In this work, we utilize the native CLIP visual and textual encoders to enable ZSTAL, without incorporating any additional adapters or fine-tuning modules.

**Zero-Shot Temporal Action Localization.** ZSTAL Ju et al. (2022); Yan et al. (2023) extends conventional closed-set TAL to the zero-shot setting, allowing models to recognize unseen actions without the need for annotated examples. EffPrompt Ju et al. (2022) utilizes an external detector to generate proposals and leverages CLIP with learnable prompts for classifying these proposals. STALE Nag et al. (2022) generates category-agnostic candidate segments and performs localization and classification simultaneously. DeTAL Li et al. (2024b) adopts a decoupled pipeline, where proposals are first generated from I3D visual features, then fused with CLIP visual features, and finally classified based on textual similarity. Recent studies Du et al. (2024); Phan et al. (2024) have further explored transformer-based backbones Meng et al. (2021); Wang et al. (2022) for ZSTAL. Building upon these advances, we adopt a transformer-based encoder-decoder architecture that unifies action detection and classification via a set of learnable queries.

**Semantic Co-modeling in Video Understanding.** Recent advances in video understanding Sun et al. (2025); Li et al. (2025a); Chen et al. (2025); Li et al. (2025c) have underscored the importance of incorporating semantics to bridge the gap between visual and textual modalities. In the ZSTAL task, mProTEA Raza et al. (2024) introduces an auxiliary training network to mitigate this gap between visual and textual features, improving localization accuracy. Ti-FAD Lee et al. (2024), built upon ActionFormer Zhang et al. (2022), performs cross-modal fusion Vaswani et al. (2017) and employs an external scoring mechanism during inference. CSP Wang et al. (2025) maps visual features into a shared embedding space and enforces semantic alignment across modalities via a loss function. However, these ZSTAL approaches limit cross-modal collaboration to a single stage of the pipeline, constraining their overall effectiveness. In contrast, our proposed **PSFTR** progressively integrates textual semantics into both the encoder and decoder stages, and further refines query representations during classification, providing consistent semantic guidance throughout the pipeline.

## 3 METHOD

### 3.1 PROBLEM DEFINITION

We formally define the ZSTAL task as follows. Given a training set of untrimmed videos $\mathcal{D}_{\text{train}}$, each video $V \in \mathcal{D}_{\text{train}}$ consists of a sequence of frames $F = \{f_t\}_{t=1}^T$, where $T$ denotes the total number of frames. Each video is annotated with a set of action instances $Y = \{(s_n, e_n, c_n)\}_{n=1}^N$, where $s_n$ and $e_n$ represent the start and end times of the $n$-th action instance, and $c_n$ denotes its category label. In the zero-shot setting, the label spaces of the training and test sets are disjoint, i.e., $\mathcal{C}_{\text{train}} \cap \mathcal{C}_{\text{test}} = \emptyset$, meaning that all action categories in $\mathcal{D}_{\text{test}}$ are unseen during training. The objective of ZSTAL is to detect and classify action segments belonging to unseen categories.

### 3.2 METHOD OVERVIEW

As shown in Fig. 2, our method consists of three key stages: an encoder, a decoder, and a classifier. These stages comprise three core components: **Semantic Synergy-Aware Encoding (SSAE)**,

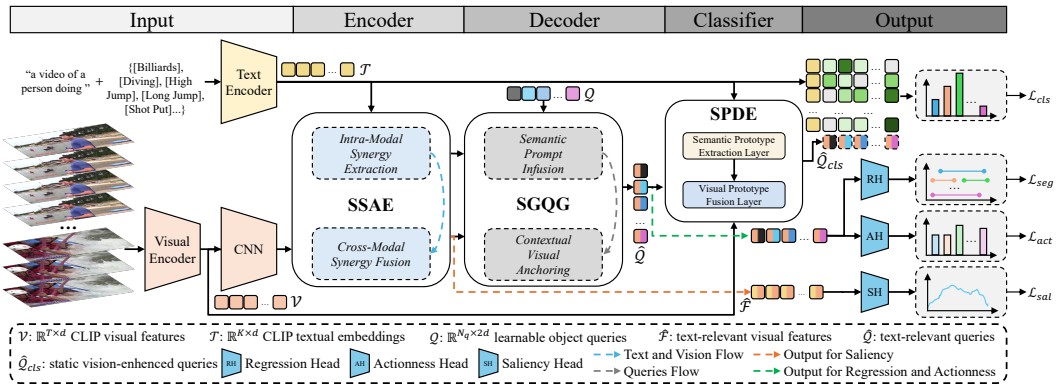

Figure 2: Overview of PSFTR. The **SSAE** module processes raw visual and textual features to produce semantically aligned outputs of both modalities. The **SGQG** module takes the outputs from the previous stage along with learnable queries as input, integrating textual semantics into the queries and then capturing visual features related to the textual semantics. Finally, the **SPDE** module injects text-relevant static visual features into the queries, enhancing their classification performance.

**Semantics-Guided Query Generation (SGQG)**, and **Semantic Prototype-Driven Enhancement (SPDE)**, as shown in Fig. 3. Specifically, the encoder enables mutual interaction between visual and textual features to achieve effective cross-modal fusion. The decoder generates queries enriched with textual semantics, allowing them to better capture text-relevant visual patterns. Classifier refines the text-aligned visual representations within the queries, improving their classification performance.

### 3.3 SEMANTIC REPRESENTATION

By leveraging CLIP Radford et al. (2021) encoders pretrained on large-scale image-text corpora, our method inherits rich cross-modal alignment, enabling strong zero-shot reasoning capabilities. Given an untrimmed video with frames $F = \{f_t\}_{t=1}^{T}$, we extract frame-level visual features using the CLIP visual encoder $v_t = \Psi_{\text{CLIP-V}}(f_t)$ and aggregate them into a sequence $\mathcal{V} = [v_1, \ldots, v_T] \in \mathbb{R}^{T \times d}$. For the $K$ candidate action categories, we construct natural-language prompts in the form of *"a video of a person doing* [ACTION]*"* to obtain $\{z_k\}_{k=1}^{K}$, which are then encoded by the CLIP text encoder as $\tau_k = \Psi_{\text{CLIP-T}}(z_k)$, resulting in the text embedding matrix $\mathcal{T} = [\tau_1, \ldots, \tau_K] \in \mathbb{R}^{K \times d}$. To further capture temporal dependencies in $\mathcal{V}$, we apply a 1D convolution and obtain $\widetilde{\mathcal{V}}$.

### 3.4 SEMANTIC SYNERGY-AWARE ENCODING

In ZSTAL, most existing methods rely solely on visual features to generate action proposals, which inevitably biases models toward the characteristics of seen actions and limits their ability to generalize to unseen categories. To overcome this limitation, we propose the **Semantic Synergy-Aware Encoding (SSAE)** module, a two-phase component integrated into the transformer encoder. In the first phase, independent multi-head self-attention is applied separately to the video features and textual embeddings, enhancing the temporal coherence of visual features while improving the semantic discriminability of textual embeddings. In the second phase, a bidirectional cross-modal attention mechanism fuses the two modalities, refining visual features with text-relevant semantics and aligning textual embeddings with action-relevant visual features. This *"intra-modal first, cross-modal later"* design produces encoder outputs that are both visually contextualized and semantically grounded, yielding discriminative representations that can generalize effectively to unseen actions.

#### 3.4.1 INTRA-MODAL SYNERGY EXTRACTION

The Intra-Modal Synergy Extraction (IMSE) strategy captures the intrinsic dependencies in visual features and textual embeddings, establishing a solid foundation for effective cross-modal alignment.

On the visual side, the temporally visual features $\widetilde{\mathcal{V}}$ are refined using a deformable self-attention (DA) module Zhu et al. (2020), which attends to a sparse set of informative temporal locations to

Figure 3: Architectures of SSAE, SGQG, and SPDE. These components progressively fuse and refine modalities across multiple stages, enhancing the model's comprehension of textual semantics.

suppress background noise, yielding contextually filtered visual features $\widetilde{\mathcal{F}}$. In parallel, the text embeddings $\mathcal{T}$ are enhanced via a multi-head self-attention (SA) mechanism that models intra-textual dependencies and captures contextual semantics among tokens, yielding a refined textual representation $\widetilde{\mathcal{T}}$ with more discriminative semantics. Together, $\widetilde{\mathcal{F}}$ and $\widetilde{\mathcal{T}}$ serve as purified features that establish a semantically enriched basis for subsequent cross-modal refinement.

### 3.4.2 CROSS-MODAL SYNERGY FUSION

To exploit the complementarity of visual and textual modalities, we propose the Cross-Modal Synergy Fusion (CMSF) strategy that performs bidirectional refinement between them. Specifically, semantic priors from textual embeddings are injected into visual features via cross-attention (CA), enabling the model to capture text-relevant visual features for unseen actions. Simultaneously, textual embeddings are dynamically conditioned on visual features, enhancing their sensitivity to visual patterns and grounding semantic representations in visual modality. This mutual refinement facilitates representational synergy between modalities, improving the model's generalization capability.

**Semantic Guidance from Text to Vision.** To inject semantic priors from textual embeddings into the visual modality, we first project the visual features and textual embeddings into query, key, and value, and then compute cross-attention weights to capture textual embeddings:

$$
\begin{aligned}
\mathcal{F}_{\mathrm{A}} &= \mathrm{CA}(\widetilde{\mathcal{F}}, \widetilde{\mathcal{T}}, \widetilde{\mathcal{T}}), \\
\widehat{\mathcal{F}} &= \mathrm{FFN}(\mathrm{LayerNorm}(\mathcal{F}_{\mathrm{A}} + \widetilde{\mathcal{F}})),
\end{aligned}
\tag{1}
$$

the attention output $\mathcal{F}_{\mathrm{A}}$ aggregates text-relevant embeddings, which are then fused with the original visual features via a residual connection followed by layer normalization. This process adaptively enhances the discriminative capacity of the visual features by emphasizing regions aligned with the textual semantics, improving the model's ability to capture visual features of unseen categories.

**Vision-to-Text Contextual Feedback.** To establish robust bidirectional modalities fusion, we project the textual embeddings and visual features into query, key, and value, and compute cross-attention from the textual embeddings to the visual modality as follows:

$$
\begin{aligned}
\mathcal{T}_{\mathrm{A}} &= \mathrm{CA}(\widetilde{\mathcal{T}}, \widetilde{\mathcal{F}}, \widetilde{\mathcal{F}}), \\
\widehat{\mathcal{T}} &= \mathrm{FFN}(\mathrm{LayerNorm}(\mathcal{T}_{\mathrm{A}} + \widetilde{\mathcal{T}})),
\end{aligned}
\tag{2}
$$

the attention output $\mathcal{T}_{\mathrm{A}}$ incorporates visual features relevant to the textual embeddings and is refined via residual fusion followed by a feed-forward network. By grounding textual semantics in the visual modality, this feedback loop enhances textual semantics and yields more precise textual embeddings.

### 3.5 SEMANTICS-GUIDED QUERY GENERATION

Traditional action boundary regression methods either predict frame-level boundaries or depend on pre-generated proposals, which biases the model toward seen actions and limits its ability to unseen actions. To overcome this issue, we propose the **Semantics-Guided Query Generation (SGQG)** module, which injects textual semantics into learnable queries, enabling them to attend to text-relevant visual features during both classification and localization. By initializing the queries with

textual semantics derived from the category, SGQG enhances their interaction with visual features, reducing the ambiguity of purely visual queries and improving the localization of unseen actions.

### 3.5.1 SEMANTIC PROMPT INFUSION.

In the Semantic Prompt Infusion (SPI) strategy, let $\mathcal{Q} \in \mathbb{R}^{N_q \times 2d}$ denote the set of learnable queries, where $N_q$ is the number of queries and $d$ is the feature dimension. After obtaining $\overline{\mathcal{Q}}$ through self-attention applied to $\mathcal{Q}[d:]$, we integrate textual semantics into the decoding process as follows:

$$\mathcal{Q}_{\mathrm{A}} = \mathrm{CA}(\overline{\mathcal{Q}}, \widehat{\mathcal{T}}, \widehat{\mathcal{T}}),$$
$$\widetilde{\mathcal{Q}} = \mathrm{FFN}(\mathrm{LayerNorm}(\mathcal{Q}_{\mathrm{A}} + \mathcal{Q})), \tag{3}$$

the resulting queries $\mathcal{Q}_{\mathrm{A}}$ are fused with the original queries via a residual connection, followed by layer normalization and a feed-forward network. This semantics-guided query mechanism modulates the queries based on textual semantics, enhancing their sensitivity to action-relevant semantics.

### 3.5.2 CONTEXTUAL VISUAL ANCHORING.

After incorporating textual semantics into the queries, the queries $\widetilde{\mathcal{Q}}$ attend to the visual features $\widehat{\mathcal{F}}$ via the Contextual Visual Anchoring (CVA) strategy, yielding the final output queries $\widehat{\mathcal{Q}}$ as follows:

$$\widetilde{\mathcal{Q}}_{\mathrm{A}} = \mathrm{DA}(\widetilde{\mathcal{Q}}, \widehat{\mathcal{F}}, \widehat{\mathcal{F}}),$$
$$\widehat{\mathcal{Q}} = \mathrm{FFN}(\mathrm{LayerNorm}(\widetilde{\mathcal{Q}}_{\mathrm{A}} + \widetilde{\mathcal{Q}})). \tag{4}$$

This dual conditioning on both textual and visual modalities improves the interpretability of the queries and strengthens the model's capability to recognize and localize actions of unseen categories.

### 3.6 SEMANTIC PROTOTYPE-DRIVEN ENHANCEMENT

Classifying proposals or learnable queries solely based on the similarity between temporally aggregated visual features and textual embeddings often dilutes the most salient moments of an action, degrading classification performance. To address this issue, we propose the **Semantic Prototype-Driven Enhancement (SPDE)** module, which integrates static visual features derived from textual semantic prototypes into the queries. This integration amplifies text-relevant features and strengthens the model's discriminative capacity, leading to more accurate classification of unseen actions.

We first compute cross-modal attention between the textual embeddings $\mathcal{T}$ and the visual features $\mathcal{V}$ for visual semantic prototype extraction, resulting in a refined visual representation $\widehat{\mathcal{V}}$:

$$\widehat{\mathcal{V}} = \mathrm{Matmul}\big(\mathrm{Softmax}(\mathrm{Matmul}(\mathcal{T}, \mathcal{V})), \mathcal{V}\big). \tag{5}$$

Global average pooling is applied to generate a compact visual summary $\widehat{\mathcal{V}}$, which is fused into the queries through a residual projection, enabling effective visual prototype fusion with the queries:

$$\widehat{\mathcal{Q}}_{\mathrm{cls}} = \widehat{\mathcal{Q}} + \mathrm{Linear}(\mathrm{MeanPool}(\widehat{\mathcal{V}})). \tag{6}$$

By integrating static visual features with dynamic queries, this module provides stronger semantic guidance, mitigates query ambiguity, and significantly improves classification performance.

### 3.7 TRAINING AND INFERENCE

**Query Matching.** Following standard query-based models Liu et al. (2022); Kim et al. (2023), we perform Hungarian matching Kuhn (1955) between predictions and ground truths. Let $\mathcal{Y} = \{y_j\}_{j=1}^{N_q}$ denote the ground-truth set and $\hat{\mathcal{Y}} = \{\hat{y}_j\}_{j=1}^{N_q}$ denote the set of predictions. The matching loss considers both classification probabilities and the distance between predicted and ground-truth action segments, formulated as:

$$\mathcal{L}_{match}(\hat{y}_j, y_j) = \mathcal{L}_{cls}(\hat{c}_j, c_j) + \mathcal{L}_{seg}(\hat{s}_j, s_j), \tag{7}$$

where $c_j$ and $s_j$ are the category label and action segment of $y_j$. The classification term $\mathcal{L}_{cls}(\hat{c}_j, c_j)$ uses the cross-entropy loss. The segmentation term $\mathcal{L}_{seg}(\hat{s}_j, s_j)$ measures the discrepancy between predicted and ground-truth segment locations, defined as:

$$\mathcal{L}_{seg}(\hat{s}_j, s_j) = \lambda_{iou} \mathcal{L}_{iou}(\hat{s}_j, s_j) + \lambda_{coord} \mathcal{L}_{L_1}(\hat{s}_j, s_j), \tag{8}$$

where $\mathcal{L}_{L_1}$ is the $L_1$ distance and $\mathcal{L}_{iou}$ is the IoU loss Rezatofighi et al. (2019), defined as one minus the IoU. The coefficients $\lambda_{iou}$ and $\lambda_{coord}$ control the relative contributions of the two terms.

**Query Actionness.** We employ the action-aware loss Du et al. (2024) to provide category-agnostic perception. Let $\mathcal{A} = \{a_j\}_{j=1}^{N_q}$ and $\hat{\mathcal{A}} = \{\hat{a}_j\}_{j=1}^{N_q}$ denote the ground-truth and predicted actionness scores for $N_q$ queries. The actionness loss is computed using the focal loss (FL) Lin et al. (2017):

$$\mathcal{L}_{act}(\hat{a}_j, a_j) = \text{FL}(\hat{a}_j, a_j). \tag{9}$$

**Frame Saliency.** To further enhance query generation, we apply a frame-level saliency loss Du et al. (2024) to align visual and textual modalities. Let $\mathcal{G} = \{g_t\}_{t=1}^T$ and $\hat{\mathcal{G}} = \{\hat{g}_t\}_{t=1}^T$ represent the ground-truth and predicted saliency maps over frames, respectively. The saliency loss is defined as:

$$\mathcal{L}_{sal}(\hat{g}_t, g_t) = \text{BCE}(\hat{g}_t, g_t), \tag{10}$$

where BCE denotes the binary cross-entropy loss.

**Overall Training Objective.** Our model is trained end-to-end to jointly optimize all components. The overall training objective is:

$$\mathcal{L}_{total} = \sum_{j=1}^{N_q} \left[ \mathcal{L}_{match}(\hat{y}_j, y_j) + \lambda_1 \mathcal{L}_{act}(\hat{a}_j, a_j) \right] + \frac{\lambda_2}{T} \sum_{t=1}^{T} \mathcal{L}_{sal}(\hat{g}_t, g_t), \tag{11}$$

where $\lambda_1$ and $\lambda_2$ are hyperparameters that balance the contributions of the respective loss terms.

### 3.7.1 INFERENCE.

During inference, we apply ROIAlign He et al. (2017) to extract feature maps corresponding to the action intervals for classification. For each query, a confidence score is calculated by combining its classification score with its actionness score, which is used to obtain the final action prediction.

## 4 EXPERIMENTS

### 4.1 EXPERIMENT SETTINGS

**Datasets.** We evaluate our method on two widely used ZSTAL benchmarks: **THUMOS14** Idrees et al. (2017) and **ActivityNet1.3** Caba Heilbron et al. (2015). THUMOS14 consists of 200 untrimmed training videos and 213 testing videos spanning 20 action categories, primarily involving sports-related activities. ActivityNet1.3 contains 19,994 untrimmed videos annotated with 200 daily activity categories. For the zero-shot setting, we follow the experimental protocols introduced in EffPrompt Ju et al. (2022). In the **50%-50%** split, the model is trained on 50% of the categories and evaluated on the remaining 50%. In the **75%-25%** split, training is performed on 75% of the categories and testing on the remaining 25%. To ensure statistical robustness, we randomly sample 10 different category splits for each setting and report the average performance across all splits.

### 4.1.1 EVALUATION METRIC.

We adopt mean Average Precision (mAP) as the primary evaluation metric, following standard practice in TAL. For THUMOS14, mAP is computed at multiple temporal Intersection over Union (tIoU) thresholds ranging from 0.3 to 0.7 with a step size of 0.1. For ActivityNet1.3, we report the mAP over tIoU thresholds from 0.5 to 0.95 with a step size of 0.05. The final mAP score is obtained by averaging the average precision (Avg) across all test splits.

### 4.1.2 IMPLEMENTATION DETAILS

For a fair comparison with previous works, we adopt only the visual and text encoders from the pre-trained CLIP (ViT-B/16) to extract video features and text prompts, with the feature dimension set to $d = 512$. The numbers of encoder and decoder layers in the transformer are set to 2 and 5 for THUMOS14, and 2 and 2 for ActivityNet1.3, respectively. We use the AdamW optimizer Loshchilov & Hutter (2017) with a batch size of 16 and a weight decay of $1 \times 10^{-4}$. The number of action queries is set to 40 for THUMOS14 and 30 for ActivityNet1.3, while the learning rates are set to $1 \times 10^{-4}$ and $5 \times 10^{-5}$, respectively. Our method is implemented in PyTorch, and all experiments are conducted on an NVIDIA GTX 4070Ti GPU. More details are provided in Appendix D.1.

Table 1: Performance Comparison with the SOTA Methods on THUMOS14 and ActivityNet1.3

| Train Split | Model | THUMOS14 | | | | | | ActivityNet1.3 | | | |
|---|---|---|---|---|---|---|---|---|---|---|---|
| | | 0.30 | 0.40 | 0.50 | 0.60 | 0.70 | Avg | 0.50 | 0.75 | 0.95 | Avg |
| 75% Seen 25% Unseen | B-II Nag et al. (2022) | 28.5 | 20.3 | 17.1 | 10.5 | 6.9 | 16.6 | 32.6 | 18.5 | 5.8 | 19.6 |
| | B-I Nag et al. (2022) | 33.0 | 25.5 | 18.3 | 11.6 | 5.7 | 18.8 | 35.6 | 20.4 | 2.1 | 20.2 |
| | EffPrompt Ju et al. (2022) | 39.7 | 31.6 | 23.0 | 14.9 | 7.5 | 23.3 | 37.6 | 22.9 | 3.8 | 23.1 |
| | STALE Nag et al. (2022) | 40.5 | 32.3 | 23.5 | 15.3 | 7.6 | 23.8 | 38.2 | 25.2 | 6.0 | 24.9 |
| | DeTAL Li et al. (2024b) | 39.8 | 33.6 | 25.9 | 17.4 | 9.9 | 25.3 | 39.3 | 26.4 | 5.0 | 25.8 |
| | mProTEA Raza et al. (2024) | 43.1 | 38.2 | 28.2 | 18.1 | 8.7 | 27.91 | 44.5 | 27.4 | **7.9** | 27.6 |
| | CSP Wang et al. (2025) | 42.7 | 35.5 | 26.4 | 18.5 | **12.0** | 27.0 | 41.1 | 28.8 | 7.4 | 28.1 |
| | PSFTR | **47.41** | **39.06** | **29.68** | **19.31** | 9.48 | **28.99** | **45.40** | **30.74** | 6.76 | **29.91** |
| 50% Seen 50% Unseen | B-II Nag et al. (2022) | 21.0 | 16.4 | 11.2 | 6.3 | 3.2 | 11.6 | 25.3 | 13.0 | 3.7 | 12.9 |
| | B-I Nag et al. (2022) | 27.2 | 21.3 | 15.3 | 9.7 | 4.8 | 15.7 | 28.0 | 16.4 | 1.2 | 16.0 |
| | EffPrompt Ju et al. (2022) | 37.2 | 29.6 | 21.6 | 14.0 | 7.2 | 21.9 | 32.0 | 19.3 | 2.9 | 19.6 |
| | STALE Nag et al. (2022) | 38.3 | 30.7 | 21.2 | 13.8 | 7.0 | 22.2 | 32.1 | 20.7 | 5.9 | 20.5 |
| | DeTAL Li et al. (2024b) | 38.3 | 32.3 | 24.4 | 16.3 | 9.0 | 24.1 | 34.4 | 23.0 | 4.0 | 22.4 |
| | mProTEA Raza et al. (2024) | **41.2** | **36.3** | **26.3** | 16.8 | 8.4 | **26.1** | **41.8** | 24.6 | **6.1** | 25.6 |
| | CSP Wang et al. (2025) | **41.2** | 33.4 | 24.8 | **17.3** | **10.9** | 25.5 | 38.4 | 26.4 | 5.2 | 25.7 |
| | PSFTR | 40.73 | 32.50 | 23.71 | 14.54 | 7.05 | 23.71 | 41.40 | **27.77** | 5.64 | **27.03** |

## 4.2 COMPARISON WITH STATE-OF-THE-ART METHODS

Tab. 1 summarizes the performance of various methods on the ZSTAL benchmark. Our proposed PSFTR demonstrates strong performance on both THUMOS14 and ActivityNet1.3. Under the **75% seen and 25% unseen** split, PSFTR achieves the highest average mAP, reaching 28.99% on THU-MOS14 and 29.91% on ActivityNet1.3, outperforming all competing approaches. In the more challenging **50% seen and 50% unseen** split, PSFTR obtains an average mAP of 23.71% on THU-MOS14, which is lower than mProTEA (26.1%) and CSP (25.5%). However, on ActivityNet1.3, PSFTR (27.03%) surpasses both mProTEA (25.6%) and CSP (25.7%). Notably, while mProTEA exhibits a substantial performance drop on ActivityNet1.3 compared to THUMOS14, PSFTR maintains stable performance. These results indicate that our method exhibits a certain degree of dependence on data scale, as the ActivityNet1.3 is larger than THUMOS14. However, they also highlight an important observation: as the dataset scale increases, PSFTR demonstrates stronger generalization capability. This improvement primarily arises from our strategy of progressively integrating textual semantics into the pipeline, including feature refinement, query decoding, and query classification, which enables the model to effectively capture the characteristics of unseen actions and to more accurately distinguish the semantics of diverse actions, exhibiting stronger generalization and achieving precise temporal action localization. Additional analysis is provided in Appendix F.1.

## 4.3 ABLATION STUDIES

**Component Analysis of Transformer.** Tab. 2 presents a comprehensive component analysis of PSFTR on the THUMOS14 dataset, evaluating the effects of three key components: text-to-vision, vision-to-text, and text-to-query. The baseline in Row 1 contains only the SGQG component. Rows 2-7 progressively explore the contribution of each module. In the encoder stage (Rows 2 and 3), incorporating text-to-vision alone or combining it with vision-to-text leads to consistent improvements over the baseline. This demonstrates that textual guidance helps the model focus on visually relevant regions by aligning visual features with textual semantics. In contrast, using only text-to-query in the decoder (Row 4) produces suboptimal performance because the queries must jointly learn visual and textual modality, which causes cross-modal misalignment and ambiguity regarding which modality should dominate. Rows 5 and 6 show that introducing either text-to-vision or vision-to-text in the encoder further boosts performance compared to the baseline. Finally, Row 7 achieves the best results by integrating both encoder fusion and text-guided query generation, where the encoder first establishes strong cross-modal alignment, providing the decoder with well-structured and semantically consistent inputs. Additional analysis is provided in Appendix F.2.

**Analysis of SPDE.** Tab. 3 presents the ablation study of the SPDE module. Without incorporating SPDE into PSFTR, the performance is 27.97%. After applying different SPDE variants: Con. + MLP, Res., and MLP + Res., the performance changes to 27.50%, 28.31%, and 28.99%, respectively. The performance drop with **Con. + MLP** is attributed to the concatenation operation increasing feature dimensionality, which complicates representation learning and thus reduces performance. In

Table 2: Ablation Study on the Impact of Encoder and Decoder Designs in Transformer.

| Row | Method | Encoder | | Decoder | mAP@Avg (%) | |
|---|---|---|---|---|---|---|
| | | text-to-vision | vision-to-text | text-to-query | 50%-50% | 75%-25% |
| 1 | Baseline | | | | 21.89 | 26.34 |
| 2 | PSFTR | ✓ | | | 22.17↑0.28 | 26.75↑0.41 |
| 3 | | ✓ | ✓ | | 22.76↑0.87 | 27.07↑0.73 |
| 4 | | | | ✓ | 21.23↓0.66 | 25.55↓0.79 |
| 5 | | ✓ | | ✓ | 22.05↑0.16 | 26.64↑0.30 |
| 6 | | | ✓ | ✓ | 22.55↑0.66 | 27.11↑0.77 |
| 7 | | ✓ | ✓ | ✓ | 23.71↑1.82 | 28.99↑2.65 |

Table 3: Ablation Study on the Impact of Fusion Strategies in SPDE.

| Fusion Type | mAP@tIoU (%) | | | |
|---|---|---|---|---|
| | 0.3 | 0.5 | 0.7 | Avg |
| w/o SPDE | 46.36 | 27.98 | 9.26 | 27.97 |
| Con. + MLP | 45.51 | 27.84 | 9.32 | 27.50 |
| Res. | 46.43 | 28.95 | 9.35 | 28.31 |
| MLP + Res. | **47.41** | **29.68** | **9.48** | **28.99** |

Table 4: Ablation Study on the Effect of Different Prompt Formats.

| Prompt | mAP@tIoU (%) | | | |
|---|---|---|---|---|
| | 0.3 | 0.5 | 0.7 | Avg |
| w/o Prompt | 44.49 | 27.65 | 8.94 | 27.12 |
| Prompt1 | 42.67 | 26.76 | 9.24 | 26.32 |
| Prompt2 | 44.94 | 27.15 | 9.41 | 27.30 |
| Prompt3 | **47.41** | **29.68** | **9.48** | **28.99** |

contrast, both **Res.** and **MLP + Res.** outperform the baseline without SPDE, demonstrating the effectiveness of SPDE in enhancing temporal action localization.

**Analysis of Prompts.** Tab. 4 presents the ablation study on different prompt formulations. **w/o Prompt** uses only the category name [ACTION] as the text input. **Prompt1** is formulated as "a video of " + [ACTION], **Prompt2** as "a video of action " + [ACTION], and **Prompt3** as "a video of a person doing " + [ACTION]. The results show that certain prompts can negatively impact performance, likely because the additional context obscures the original semantic meaning of [ACTION], degrading accuracy. Among the tested variants, **Prompt3** achieves the best performance.

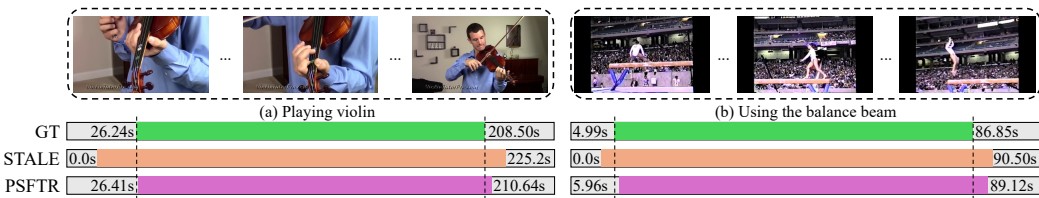

Figure 4: Comparison of visualizations of results on the ActivityNet1.3 with a 75%-25% split.

## 4.4 QUALITATIVE RESULTS

Fig. 4 shows the visualization results of our method. Green indicates the ground truth, Orange represents the results of STALE, and Pink denotes the predictions of PSFTR. As illustrated, PSFTR produces more accurate action localization than STALE, owing to the guidance provided by textual semantics. Additional qualitative analyses are included in Appendix F.4.

## 5 CONCLUSION

In this paper, we propose **PSFTR** (*Progressive Semantic Fusion TRansformer*), a novel transformer-based method for ZSTAL. PSFTR progressively strengthens the fusion between visual and textual modalities by integrating a semantic fusion transformer with a text-guided query enhancement mechanism, improving the model's ability to comprehend text-relevant actions. Our approach facilitates the capture of critical visual features for unseen actions, leading to more accurate temporal localization in ZSTAL tasks. Extensive experiments on the THUMOS14 and ActivityNet1.3 demonstrate that PSFTR generalizes effectively to unseen actions and outperforms the state-of-the-art methods.

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

## A    REPRODUCIBILITY STATEMENT

The code referenced in this work is available in the works Liu et al. (2022); Du et al. (2024). The datasets used in our experiments are widely adopted benchmarks, including THUMOS14 Idrees et al. (2017) and ActivityNet1.3 Caba Heilbron et al. (2015). The visual-language model CLIP can be accessed and reproduced following the details provided in publication Radford et al. (2021).

## B    DECLARATION OF LARGE LANGUAGE MODELS

Large Language Models (LLMs) were used solely to assist in drafting and polishing the text; all scientific content and interpretations remain the authors' own.

## C    RELATED WORK

### C.1    OVERVIEW OF ZERO-SHOT LEARNING

Zero-shot learning enables a model to generalize to novel concepts or categories at inference without any annotated training examples, by transferring semantic knowledge from seen to unseen. Recent advances in large-scale Vision-Language pretraining (ViL), exemplified by CLIP Radford et al. (2021) and ALIGN Jia et al. (2021), have achieved remarkable zero-shot performance across a wide range of image understanding domains Qian et al. (2025). These models construct a unified embedding space where visual inputs and textual descriptions are directly aligned, enabling open-vocabulary tasks via natural-language prompts without the need for task-specific fine-tuning. Building on these successes, recent research has extended ViL architectures to open-world video understanding tasks. In particular, CLIP's text encoder provides semantically rich representations that are intrinsically aligned with its visual counterpart, eliminating the reliance on hand-crafted taxonomies. Leveraging this alignment, most ZSTAL methods employ CLIP to embed both video and text inputs into a shared latent space, enabling the classification and localization of unseen actions.

## D    MODEL AND TRAINING HYPERPARAMETERS

### D.1    CONFIGURATION PARAMETERS

Table 5: Configuration Parameters in THUMOS14 and ActivityNet1.3

| Parameter | THUMOS14 | ActivityNet1.3 |
|---|---|---|
| num_classes | **20** | **200** |
| slice_size | 128 | 128 |
| slice_overlap | 0.75 | 0.75 |
| inference_slice_overlap | 0.25 | 0.25 |
| enc_layers | 2 | 2 |
| dec_layers | **5** | **2** |
| num_queries | 40 | 30 |
| ROIalign_size | 16 | 16 |
| cls_loss_coef | 2 | 2 |
| seg_loss_coef | 5 | 5 |
| iou_loss_coef | 2 | 2 |
| focal_alpha | 0.25 | 0.25 |
| gamma | 2 | 2 |
| actionness_loss_coef | 3.0 | 3.0 |
| salient_loss_coef | **3.0** | **2** |
| salient_loss_impl | BCE | BCE |
| epochs | **30** | **20** |
| lr | $1 \times 10^{-4}$ | $5 \times 10^{-5}$ |
| lr backbone | $1 \times 10^{-5}$ | $1 \times 10^{-2}$ |

Tab. 5 summarizes the key configuration parameters for CLIP zero-shot training on THUMOS14 and ActivityNet1.3. Most parameters, including slice_size, slice_overlap, inference_slice_overlap, ROIalign_size, and loss weights, are kept consistent to ensure comparability. Differences mainly reflect dataset characteristics: THUMOS14 uses a deeper decoder (5 layers) and higher salient loss weight (3.0) to better exploit limited data, while ActivityNet1.3 adopts fewer decoder layers (2 layers), a lower salient loss weight (2), and a larger backbone learning rate to accommodate more categories (200) and prevent overfitting. Training epochs and learning rates are also adjusted accordingly, balancing convergence speed, stability, and generalization. Overall, this configuration demonstrates a principled approach to adapting model capacity and optimization to dataset scale and complexity while maintaining architectural consistency.

# E ADDITIONAL EXPERIMENTS

## E.1 EFFECT OF ATTENTION HEADS

Table 6: Ablation Study on the Effect of Attention Heads in SA, CA, and DA.

| n_heads | mAP@tIoU (%) | | | | | |
|---|---|---|---|---|---|---|
| | 0.3 | 0.4 | 0.5 | 0.6 | 0.7 | Avg |
| 1 | 39.66 | 31.32 | 21.74 | 13.17 | 5.62 | 22.30 |
| 2 | 42.36 | 34.17 | 25.59 | 15.53 | 7.28 | 24.99 |
| 4 | 46.00 | 36.70 | 26.75 | 16.75 | 8.08 | 26.86 |
| 8 | **47.41** | **39.06** | **29.68** | **19.31** | **9.48** | **28.99** |
| 16 | 46.00 | 37.21 | 27.00 | 17.27 | 8.67 | 27.23 |

Tab. 6 reports the effect of varying the number of attention heads in the SA, CA, and DA modules. Overall, increasing the number of heads from 1 to 8 consistently improves performance across all tIoU thresholds, with the average mAP rising from 22.30% to 28.99%. This indicates that multi-head attention effectively captures richer temporal and cross-channel dependencies. However, further increasing the heads to 16 leads to a slight performance drop, particularly at higher tIoU thresholds (0.6 and 0.7), suggesting that excessive heads may disperse the attention or introduce instability. Notably, 8 heads achieve the best balance between precision and robustness, providing strong improvements at both low and high tIoU thresholds, and thus represent the optimal configuration.

## E.2 ANALYSIS OF CLIP ENCODER

Table 7: Comparison of Different CLIP Encoders

| Model | Patch Size | Layers | Embedding Dim | Vision Parameters | Text Parameters |
|---|---|---|---|---|---|
| ViT-B/32 | 32×32 | 12 | 512 | 87.8M | 37.8M |
| ViT-B/16 | 16×16 | 12 | 512 | 86.2M | 37.8M |
| ViT-L/14 | 14×14 | 24 | 768 | 304.0M | 85.0M |

Tab. 7 and Tab. 8 jointly present the impact of different CLIP visual encoders on ZSTAL performance using the THUMOS14 dataset. Tab. 7 summarizes the architectural differences among three CLIP variants, including patch size, number of transformer layers, embedding dimensionality, and parameter count. Among them, ViT-L/14 has the largest model capacity (304M parameters) and the finest patch resolution ($14 \times 14$). In contrast, ViT-B/32 and ViT-B/16 share the same hidden dimension of 512, with parameter counts of 87.8M and 86.2M, respectively, but differ in patch size.

Tab. 8 reports the localization performance under both the **75% seen and 25% unseen** and **50% seen and 50% unseen** splits. Across both settings, ViT-L/14 achieves the highest average mAP, benefiting from its larger capacity and finer-grained spatial modeling, which enable it to capture detailed visual features more effectively. In contrast, ViT-B/32 exhibits the lowest performance, indicating that its coarser patch size ($32 \times 32$) limits its ability to represent the fine-grained temporal features crucial for accurate action localization. ViT-B/16 performs comparably to ViT-L/14,

Table 8: Performance Comparison with Different CLIP Encoders on THUMOS14

| Train Split | Model | THUMOS14 | | | | | |
|---|---|---|---|---|---|---|---|
| | | 0.30 | 0.40 | 0.50 | 0.60 | 0.70 | Avg |
| 75% Seen 25% Unseen | ViT-B/32 | 41.87 | 34.73 | 25.16 | 15.03 | 7.07 | 24.77 |
| | ViT-B/16 | 47.41 | 39.06 | 29.68 | 19.31 | 9.48 | 28.99 |
| | ViT-L/14 | **47.90** | **39.48** | **29.72** | **18.82** | **9.51** | **29.08** |
| 50% Seen 50% Unseen | ViT-B/32 | 36.23 | 29.11 | 20.82 | 12.67 | 5.76 | 20.92 |
| | ViT-B/16 | 40.73 | 32.50 | 23.71 | 14.54 | 7.05 | 23.71 |
| | ViT-L/14 | **42.77** | **33.92** | **24.67** | **15.25** | **7.32** | **24.79** |

achieving 28.99% mAP on the 75%-25% split and 23.71% on the 50%-50% split, while maintaining a substantially smaller model size. Although ViT-L/14 outperforms ViT-B/16, for a fair comparison with previous methods, only the results using ViT-B/16 are reported in Tab. 1.

Overall, these results indicate that reducing the patch size from 32 to 16 significantly boosts ZSTAL performance, even without increasing model depth or parameter count. Thus, ViT-B/16 provides an effective balance between computational efficiency and localization accuracy for ZSTAL tasks.

## F MORE VISUALIZATION ANALYSIS

### F.1 ANALYSIS OF MAIN RESULTS

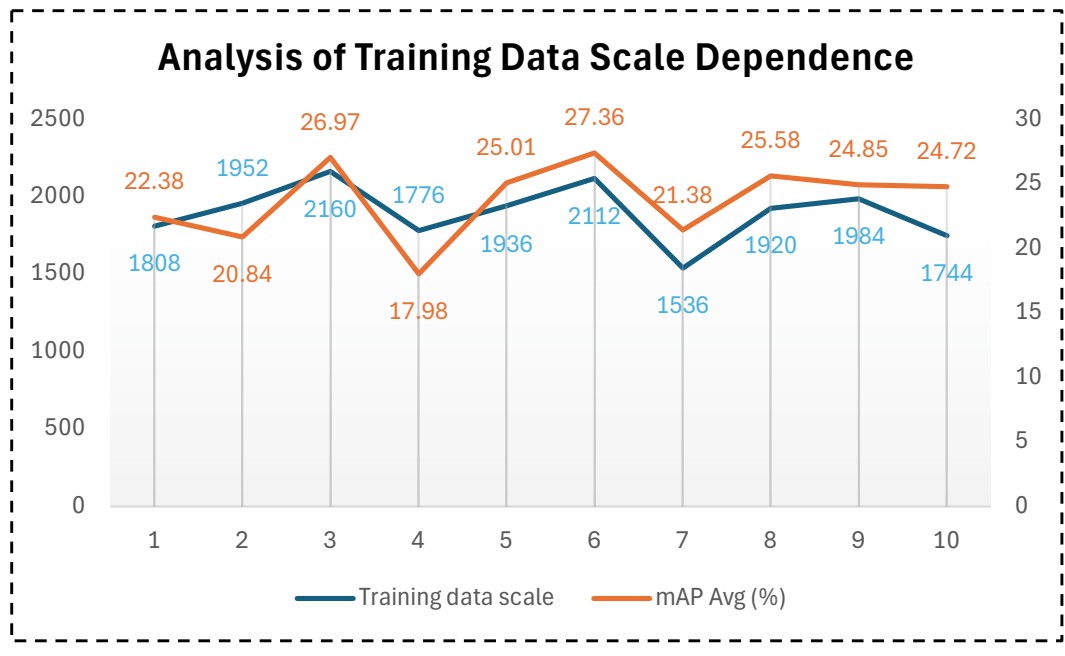

Figure 5: Training data scale and Mean Average Precision (mAP) under 10 different 50%-50% category splits on the THUMOS14 dataset

Fig. 5 illustrates the impact of training data scale on mAP performance across 10 distinct 50%-50% category splits on the THUMOS14 dataset. The results reveal a clear positive correlation: mAP steadily increases as the number of training videos grows across nearly all splits, except for Split 1. This trend highlights the crucial role of abundant training data in enabling the model to learn robust and transferable representations that generalize effectively to unseen categories. While semantic overlap between seen and unseen categories may also influence mAP, our quantitative analysis shows

that expanding the training dataset consistently yields performance improvements. These findings underscore the pivotal importance of training data scale in advancing ZSTAL performance.

## F.2 ANALYSIS OF COMPONENTS

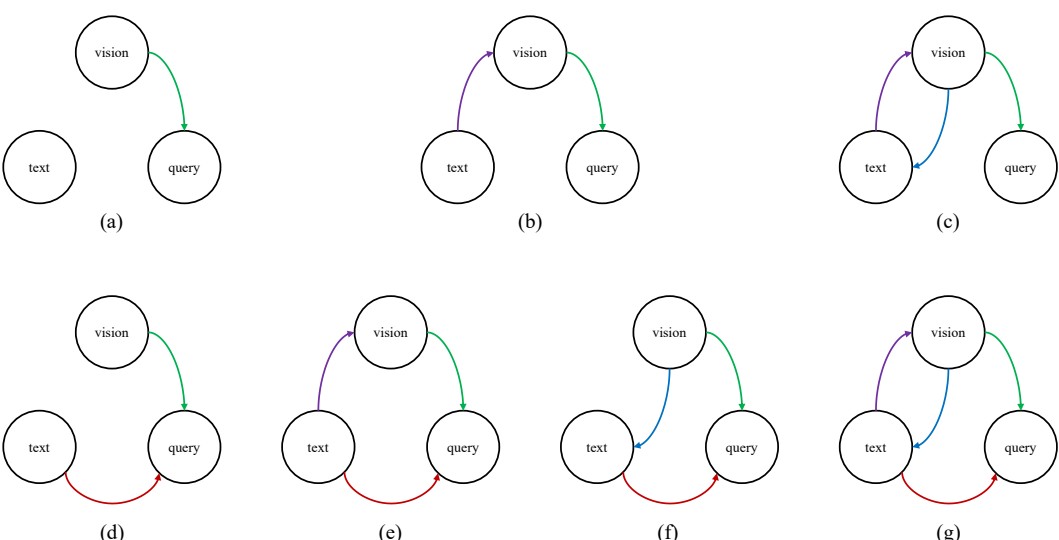

Figure 6: Data flow of PSFTR under different module configurations. (a) The standard transformer relies solely on visual features, ignoring textual information. (b-c) Gradually incorporating textual guidance into visual feature learning enhances semantic complementarity, leading to notable performance gains (Rows 2-3 in Tab. 2). (d) Directly combining both modalities without proper alignment can introduce semantic noise, causing confusion and degrading performance (Row 4). (e-g) Introducing cross-modal pathways to align the two modalities effectively mitigates misalignment issues, yielding consistent performance improvements (Rows 5-7).

Fig. 6 illustrates the data flow of PSFTR under different module configurations before and after integration into the transformer. In the standard transformer architecture (Fig. 6 (a)), the query solely learns action appearance and motion cues from visual inputs and directly predicts the corresponding action category and temporal segment, completely ignoring textual information. Fig. 6 (b) corresponds to Row 2 in Tab. 2, where text guidance is introduced to help the model capture visual features that are semantically aligned with the text, thereby narrowing the gap between visual and textual modalities. In Fig. 6 (c) (Row 3), the visual and textual modalities complement each other's semantics, leading to a more substantial performance gain as reported in Tab. 2. Fig. 6 (d-g) correspond to Rows 4-7, respectively. The performance drop observed in Row 4 can be attributed to the configuration in Fig. 6 (d), where the query simultaneously learns from both modalities without proper alignment. This misalignment introduces semantic noise, causing the query to become confused about which modality to prioritize, ultimately degrading performance. In contrast, the designs in Fig. 6 (e-g) incorporate text-to-vision or vision-to-text pathways that effectively align the two modalities, mitigating semantic conflicts and improving overall performance.

## F.3 ANALYSIS OF DIFFERENT SPLITS

Fig. 7 reports the mAP results under 10 different splits of seen and unseen categories on the THU-MOS14 dataset, with two ratio settings: **75%-25%** and **50%-50%**. The results reveal three key observations. First, the 75%-25% setting achieves the highest overall mAP (35.07% in Split 6), demonstrating that when seen categories dominate and share strong semantic correlations with unseen categories, the model can effectively transfer discriminative knowledge to unseen categories. Second, the performance of the 75%-25% setting exhibits substantial fluctuations, ranging from 17.98% to 35.07%. This indicates that the learned features tend to overfit the seen categories and fail to generalize when the unseen categories differ significantly from the seen ones. Finally, the

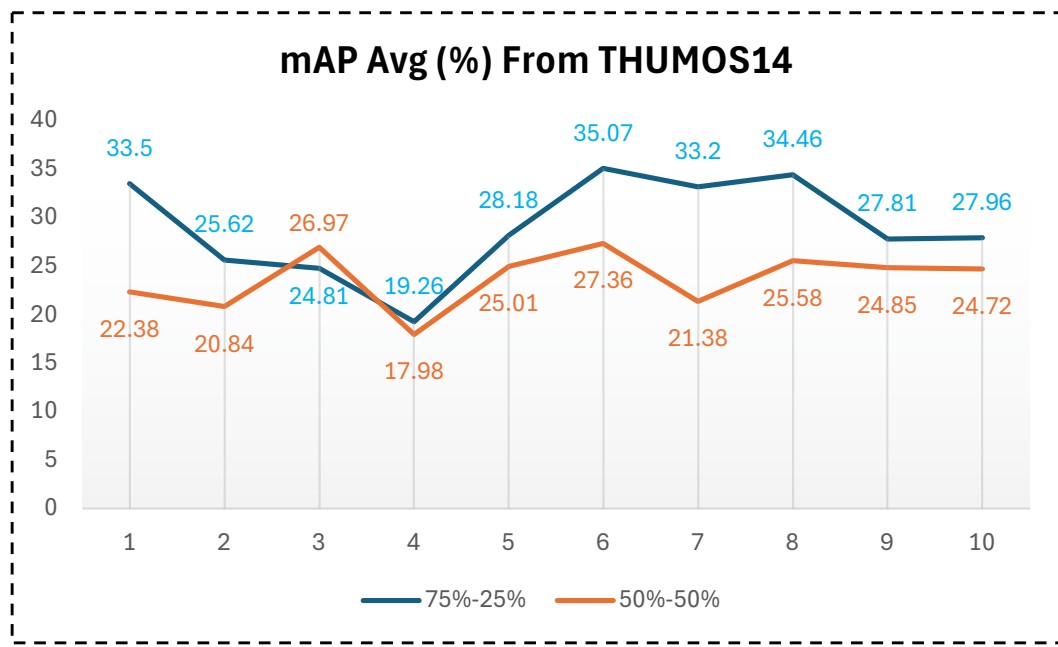

Figure 7: Mean Average Precision (mAP) under 10 different seen/unseen category splits on the THUMOS14 dataset, comparing two ratio settings (75%-25% and 50%-50%).

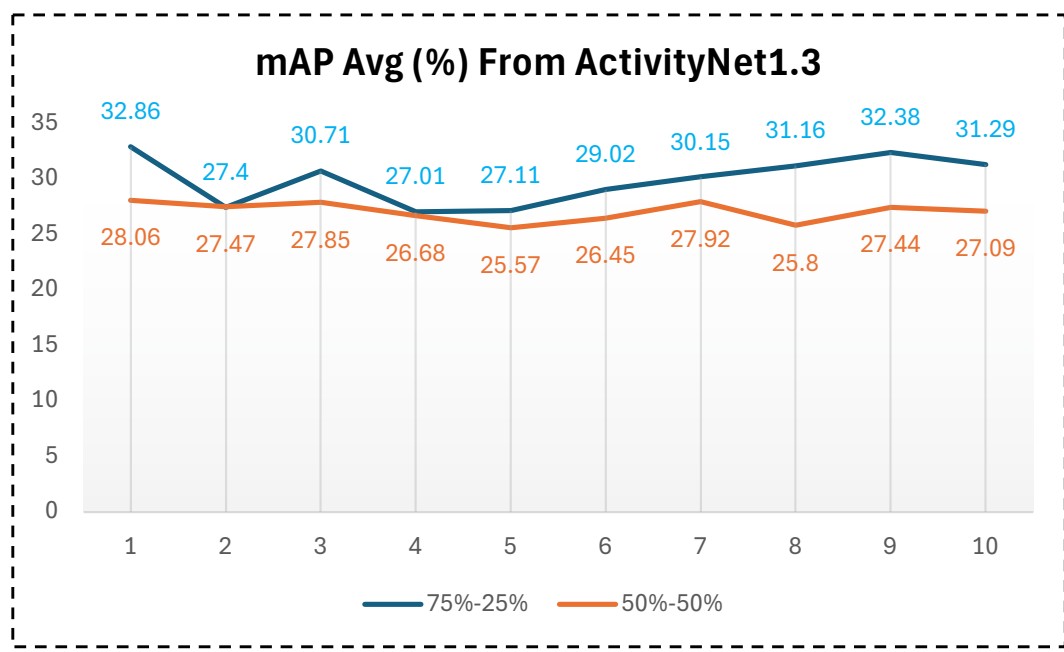

Figure 8: Mean Average Precision (mAP) under ten different seen/unseen category splits on the ActivityNet1.3 dataset, comparing two ratio settings (75%-25% and 50%-50%).

50%-50% setting delivers more stable performance, with a narrower range of 20.84%-34.46%. Although its peak performance is slightly lower than that of the 75%-25% setting, its consistency across challenging splits suggests that it provides better stability when the semantic overlap between seen and unseen categories is limited.

Fig. 8 summarizes the results across 10 different seen and unseen splits on the ActivityNet1.3 dataset. Unlike the trends observed on THUMOS14, the **75%-25%** configuration consistently outperforms the **50%-50%** configuration across nearly all splits, achieving a peak mAP of 32.86% and a minimum of 27.01%. This narrow performance range of 5.85% indicates that the 75%-25% setting is not only more accurate but also significantly more stable. In contrast, the 50%-50% configuration exhibits a relatively flat trend, with mAP values fluctuating between 25.57% and 28.06%. These findings suggest that, for large-scale and highly diverse datasets, a higher proportion of seen categories provides richer prior knowledge, which in turn enhances both the stability and generalization capability of the model.

Our results (Fig. 7 and Fig. 8) reveal a clear trade-off between accuracy, stability, and generalization across different seen and unseen splits. On ActivityNet1.3, the **75%-25%** configuration consistently achieves higher and more stable mAP scores (27.01%-32.86%), indicating that a dominant proportion of seen categories provides richer prior knowledge and strengthens model robustness. In contrast, on THUMOS14, although the **75%-25%** configuration attains the highest peak performance (35.07%), it exhibits substantial fluctuations, particularly when unseen categories are semantically distant from the seen categories, suggesting a heightened risk of overfitting. The **50%-50%** configuration, while slightly lower in peak accuracy, delivers more stable performance across challenging splits. These findings collectively suggest that our model generalizes effectively, offering both stable and accurate performance in large-scale and diverse action localization scenarios.

## F.4 VISUALIZATION OF PSFTR MODULE ABLATION

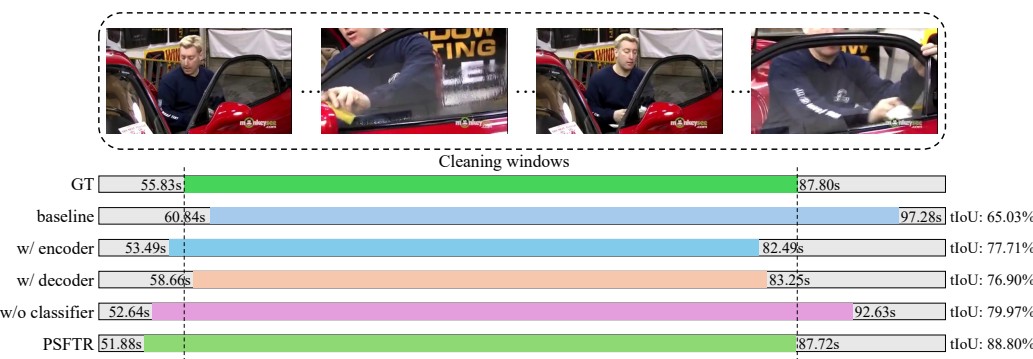

Figure 9: Ablation study of the PSFTR module. Variants include text guidance added to the encoder (**w/ encoder**), the decoder (**w/ decoder**), or both, as well as removing the SPDE module (**w/o classifier**). Visualizations and quantitative results demonstrate that adding text guidance improves boundary localization, with the full PSFTR method achieving the highest tIoU of 88.80%.

Fig. 9 provides a visual comparison of the PSFTR module ablation study. The **baseline** corresponds to a standard transformer method equipped only with the SPDE module. The variants include: **w/ encoder**, which integrates text guidance into the encoder; **w/ decoder**, which introduces text guidance into the decoder; and **w/o classifier**, which removes the SPDE module while retaining text guidance in both the encoder and decoder. As shown in Tab. 2 and Tab. 4, adding text guidance to either the encoder or decoder individually yields performance gains over the baseline. Incorporating text guidance into both the encoder and decoder further enhances performance, as evidenced by the visualizations: the model with joint guidance can more accurately localize action boundaries, with tIoU improving from 77.71% and 76.90% to 79.97%. When the full PSFTR method is employed, localization performance is further boosted to 88.80%, clearly demonstrating the effectiveness of the proposed modules.

