# OpenReview forum: "Progressive Semantic Fusion Transformer for Zero-Shot Temporal Action Localization"
_ICLR.cc/2026/Conference — ICLR 2026 Conference Withdrawn Submission_

### Official Review · Reviewer_dih9 · 2025-10-31

**Soundness:** 2
**Presentation:** 2
**Contribution:** 2
**Rating:** 4
**Confidence:** 2

**Summary:**

The paper introduces the Progressive Semantic Fusion TRansformer (PSFTR) for Zero-Shot Temporal Action Localization (ZSTAL).
It appears that the proposed method extends DETR-like architectures by injecting text semantics into three parts of the model:
(1) a Semantic Self-Attention Encoder (SSAE)
(2) a Semantic-Guided Query Generator (SGQG)
(3) a Semantic Prototype-Driven Enhancer (SPDE)
Results on THUMOS14 and ActivityNet-1.3 show modest gains (~1–2% mAP) over prior ZSTAL baselines.

**Strengths:**

1. Semantic features are consistently integrated across encoder, decoder, and classifier, forming a coherent pipeline.
2. The paper explores different prompt templates and SPDE fusion strategies, showing measurable effects.
3. Clear details on training, hyperparameters, and architecture allow reproduction.

**Weaknesses:**

I have three major concerns:
1.  The evaluation framework is rather weak -  A major evaluation gap lies due to non-inclusion of the Charades-STA and HACS benchmarks.
2. The proposed method mainly extends existing DETR-based ZSTAL frameworks by inserting CLIP text tokens; it introduces no new learning objective.
3. Since CLIP text embedding has been used in the proposed method  I wonder how this is affecting the proposed method since CLIP poorly models verbs; action semantics that rely on motion (e.g., “opening vs closing”) are semantically ambiguous.

**Questions:**

Is there any possibility of using a text encoder with better understanding of the "verb"?

---

### Official Review · Reviewer_c1Zh · 2025-11-01

**Soundness:** 2
**Presentation:** 3
**Contribution:** 2
**Rating:** 4
**Confidence:** 5

**Summary:**

This paper focuses on the task of zero-shot temporal action localization, aiming to classify and localize action instances from unseen categories in videos. One key observation is that existing work produces action proposals without introduce text for interaction. Progressive Semantic Fusion TRansformer is proposed to mitigate the limitation. Experiments are carried out on THUMOS14 and ActivityNet1.3.

**Strengths:**

[+] The manuscript is well written, with clear logics.

[+] The symbol definitions are clear, and the image visualization is complete.

[+] Many ablation experiments are conducted to analyze the effectiveness of each component.

**Weaknesses:**

[-] Introducing text for interaction during the production of action proposals can slow down the training/inference speed. The reviewer suggests comprehensive comparisons between this paper and existing methods from the perspective of efficiency (such as throughput, RT, GFLOPs)

[-] THUMOS14 and ActivityNet1.3 are both small-scale datasets, which means that there have no reference value in real-world scenarios. For zero-shot settings, one simple approach to proving generalization is to evaluate across datasets, such as training on ActivityNet1.3 and testing on THUMOS14. The reviewer suggests providing this cross-dataset performance evaluation to better understand the significance of this work to the community.

[-] In recent years, it has become a trend to unify temporal action localization and video temporal grounding in the form of multimodal large models. Some studies have demonstrated strong performance and generalization across closed-set, zero-shot, few-shot, and open-set scenarios. Please make comprehensive comparisons in terms of performance, generalization, efficiency, and practicality.

Universal Video Temporal Grounding with Generative Multi-modal Large Language Models. NeurIPS 2025

**Questions:**

Please see weaknesses.

---

### Official Review · Reviewer_6YMm · 2025-11-02

**Soundness:** 3
**Presentation:** 3
**Contribution:** 2
**Rating:** 2
**Confidence:** 5

**Summary:**

This paper proposes PSFTR, a progressive semantic fusion transformer for zero-shot temporal action localization (ZSTAL). The method injects textual embeddings at three stages—encoder, decoder, and classifier—with the claim that “semantic guidance throughout the pipeline” improves cross-modal alignment and generalization to unseen classes. Experiments are conducted on THUMOS14 and ActivityNet-1.3.

**Strengths:**

- The motivation is clearly stated and the framework is modular and easy to follow.
- The idea of incorporating text beyond a final similarity stage is reasonable to ZSTAL.

**Weaknesses:**

- Limited novelty: the method is largely a staged combination of existing components. The three modules reuse standard mechanisms (cross-attention, deformable attention, prototype fusion) and primarily rearrange them in sequence. There is no fundamentally new learning principle, objective, or semantic reasoning mechanism. The level of contribution appears incremental.
- The core claim (“semantic guidance throughout the pipeline”) is not actually demonstrated. Text embeddings are static CLIP vectors, not dynamically updated. The model uses repeated conditioning rather than true progressive semantic reasoning. No diagnostic analysis (feature visualization, probing, attention shift, semantic alignment metrics, etc.) is provided, so it remains unproven that text semantics meaningfully reshape visual representations.
- Experimental evidence is not convincing: the method fails on the more complex dataset. THUMOS14 is widely considered the more challenging benchmark (denser action instances, higher temporal ambiguity), yet the proposed method underperforms recent baselines on this dataset. If semantic fusion truly improved fine-grained localization and unseen-class reasoning, we would expect the opposite trend. This discrepancy weakens the central claim.
- Performance deteriorates under stricter IoU evaluation, contradicting the claimed benefit. Table 1 shows that the proposed method is competitive at loose IoU thresholds (0.30–0.40) but loses more clearly at stricter thresholds (0.60–0.70), and the margin vs. the strongest baseline widens as IoU increases. This indicates that the method improves coarse recall rather than precise boundary localization, directly conflicting with the claim of “enhanced fine-grained semantic guidance.”
- No computational or efficiency analysis is provided. The architecture adds three cross-modal fusion stages, which likely increase FLOPs, parameters, and memory, but no complexity comparison is reported. It is unclear whether the modest gains justify the extra cost.

**Questions:**

- Can you provide FLOPs, inference time, and parameter counts vs. baselines?
- Can you show evidence that semantic fusion actually changes visual representations across stages?
- Is a single fusion stage nearly as good as three? (Ablation necessary to justify “progressive.”)
- Why does the method lag behind baselines specifically on THUMOS14 and high-IoU evaluation?

---

### Official Review · Reviewer_MJwk · 2025-11-03

**Soundness:** 3
**Presentation:** 3
**Contribution:** 3
**Rating:** 6
**Confidence:** 4

**Summary:**

This paper addresses the task of Zero-Shot Temporal Action Localization (ZSTAL), which involves identifying and temporally localizing actions in untrimmed videos, including categories that the model has never seen during training. The authors observe that existing ZSTAL methods suffer from three main issues: weak cross-modal alignment between text and vision, visually driven proposal generation that ignores semantics, and loss of discriminative detail during feature aggregation.

To address these challenges, the paper proposes the Progressive Semantic Fusion Transformer (PSFTR). The key idea is to progressively inject textual semantics derived from a pretrained CLIP model into multiple stages of the transformer instead of fusing them only at the end. This design allows the model to maintain semantic alignment and text-awareness throughout its processing pipeline. The three main modules work together to refine visual and textual representations, improving generalization to unseen action categories.

The proposed method is evaluated on THUMOS14 and ActivityNet1.3, where it consistently outperforms previous zero-shot baselines. Ablation studies confirm that each module contributes to performance gains.

**Strengths:**

- The paper’s main strength is in the progressive fusion of textual semantics and visual features across multiple stages of the transformer architecture. This design goes beyond previous methods that only perform fusion at the final stage, making the model more semantically aware throughout its processing pipeline.
- The work demonstrates originality through the idea of using semantics-guided queries and semantic prototype enhancement, which together help the model generalize to unseen actions. The experimental results are strong, with consistent improvements over state-of-the-art methods on standard benchmarks. The paper also includes comprehensive ablation studies that clearly show the contribution of each module.
- In terms of clarity, the paper is generally well organized, with intuitive figures and a logical flow that makes the complex architecture understandable.

**Weaknesses:**

- Evaluation is confined to standard zero-shot splits on THUMOS14 and ActivityNet1.3. The work does not explore cross-dataset or out-of-distribution data, which would strengthen the claim of broad zero-shot capability.
- Only mean Average Precision (mAP) is reported. Important aspects such as temporal boundary accuracy, localization precision/recall trade-offs, and calibration or ranking consistency are not analyzed. This weakens the claim of improved temporal localization quality.
- The ablations show each module’s contribution, but the paper could include more diagnostic insight into failure cases (e.g., unseen classes with motion ambiguity or long-duration segments).
- Prompt design is briefly explored, but a more systematic study of linguistic variation (e.g., paraphrasing, template diversity) would strengthen claims about semantic fusion robustness.
- The paper does not analyze how performance varies with semantic distance between seen and unseen classes; such analysis would clarify the robustness of semantic transfer.
- Finally, the presentation of the fusion process could be made clearer for readers less familiar with attention-based architectures.

**Questions:**

- Have you evaluated or considered evaluating PSFTR in cross-dataset or out of distribution zero-shot settings (e.g., training on ActivityNet, testing on THUMOS)? Such experiments would clarify whether the fusion generalizes beyond seen-domain distributions.
- Since mAP aggregates both classification and boundary effects, could you provide complementary metrics such as temporal IoU completeness, boundary deviation, or segment precision/recall to support the claim of improved localization quality?
- What types of unseen actions remain most challenging for PSFTR (e.g., long-duration, motion-ambiguous, or semantically distant classes)? Any insights from qualitative inspection or error clustering would be valuable.
- Have you analyzed how PSFTR performs as the semantic distance between seen and unseen classes increases? For instance, do semantically closer unseen categories benefit more from the progressive fusion?

---

### Note · Authors · 2025-11-26

**Comment:**

We would like to express our sincere gratitude for the valuable feedback and suggestions provided on our manuscript. After carefully considering the reviewers' comments, we recognize that there are areas in our paper that require further enhancement to achieve a higher quality standard. To address these concerns, we have decided to withdraw our current submission. We deeply appreciate the guidance and support from the editorial team and reviewers.

**Withdrawal Confirmation:**

I have read and agree with the venue's withdrawal policy on behalf of myself and my co-authors.